# CTLA-4Ig Improves Hyperalgesia in a Mouse Model of Osteoporosis

**DOI:** 10.3390/ijms21249479

**Published:** 2020-12-13

**Authors:** Nobuto Nagao, Hiroki Wakabayashi, Gaku Miyamura, Sho Kato, Yohei Naito, Akihiro Sudo

**Affiliations:** Department of Orthopaedic Surgery, Mie University Graduate School of Medicine, 2-174 Edobashi, Tsu 514-8507, Japan; n-nagao@clin.medic.mie-u.ac.jp (N.N.); glak1225@clin.medic.mie-u.ac.jp (G.M.); katosho@clin.medic.mie-u.ac.jp (S.K.); yo-yo@clin.medic.mie-u.ac.jp (Y.N.); a-sudou@clin.medic.mie-u.ac.jp (A.S.)

**Keywords:** osteoporosis, pain, abatacept

## Abstract

This study aimed to evaluate skeletal pain associated with osteoporosis and to examine the inhibitory effects of cytotoxic T lymphocyte-associated antigen-4Ig (CTLA-4Ig) administration in ovariectomized (OVX) mice. Eight-week-old female ddY mice were assigned to three groups: sham-operated mice (SHAM) treated with vehicle, OVX mice treated with vehicle (OVX), and OVX mice treated with CTLA-4Ig (CTLA-4Ig). Vehicle or CTLA-4Ig was injected intraperitoneally, starting immediately after surgery. After 4 weeks of treatment, mechanical sensitivity was examined, and the bilateral hind limbs were removed and evaluated by micro-computed tomography, immunohistochemical analyses, and messenger RNA expression analysis. Ovariectomy induced bone loss and mechanical hyperalgesia in the hindlimbs. CTLA-4Ig treatment prevented bone loss in the hindlimbs compared to vehicle administration in the OVX group. Moreover, mechanical hyperalgesia was significantly decreased in the CTLA-4Ig treatment group in comparison to the OVX group. The expression levels of tumor necrosis factor-α (TNF-α) and sclerostin (SOST), as well as the number of osteoclasts, were increased, and the expression level of Wnt-10b was decreased in the OVX group compared with the SHAM group, whereas these parameters were improved in the CTLA-4Ig group compared with the OVX group. The novelty of this research is that CTLA-4Ig administration prevented bone loss and mechanical hyperalgesia induced by ovariectomy in the hindlimbs.

## 1. Introduction

Rheumatoid arthritis (RA) is a systemic autoimmune disease in which proinflammatory cytokines act as mediators of synovial inflammation, with resulting progressive inflammatory polyarthritis. One of the most deleterious effects is bone loss induced by the RA inflammation [1]. Osteoporosis is recognized as a major comorbidity in RA, and can result in estimated double risk of pathological fractures [2]. Bone fragility in RA patients results from a mix of systemic inflammation, circulating autoantibodies, and proinflammatory cytokine secretion that collectively have deleterious effects on bones. Systemic bone loss occurs in nearly 60% of patients with early RA, and it is also a strong predictor of radiographic joint damage [3]. RA patients typically have low bone mass at the start of their disease, indicating that bone damage already occurs before clinical inflammation starts [4].

Over the last 15 years, better knowledge of the cytokine network involved in RA enabled the development of potent inhibitors of the inflammatory process, which are classified as biologic disease-modifying antirheumatic drugs (bDMARDs) [5]. The development of bDMARDs in the late 1990s has dramatically improved the management of RA. These new drugs are very effective in the inhibition of inflammation, but there are only a few studies regarding their role in bone protection. Clinically, studies with tumor necrosis factor (TNF)-blocking agents show preservation or increase in spine and hip bone mineral density (BMD) and a better bone marker profile. Treatment with biologic drugs is associated with a decrease in bone loss [5].

Abatacept (ABT) is a recombinant fusion protein containing components of immunoglobulin G (IgG) and cytotoxic T-lymphocyte-associated protein-4 that inhibit costimulatory signals from antigen-presenting cells and prevent activation of T cells [6]. ABT, i.e., CTLA-4Ig, has been approved for the treatment of RA [7]. CTLA4-Ig significantly ameliorated signs and symptoms, improved physical function, and retarded the radiological progression of structural damage of affected joints in patients with RA. CTLA-4Ig is currently indicated for the treatment of moderate-to-severe RA resistant to methotrexate or TNF antagonists [8]. A previous prospective comparative study investigated the effects of ABT and other bDMARDs on BMD and bone metabolism markers in RA patients and revealed the effects of ABT on bone metabolism [9]. This study showed that ABT increases BMD at the femoral neck and maintains the lumbar spine, offering comparable efficacy to other bDMARDs.

Our previous studies using an animal model of ovariectomy (OVX)-induced osteoporotic pain suggested that treatment of osteoporosis is useful for osteoporotic pain [10,11]. The objective of the current study was to investigate the bone structure and pain-related behavior in a mouse model of OVX-induced osteoporosis to evaluate the effects of CTLA-4Ig.

## 2. Results

### 2.1. Measurement of Pain-Related Behavior with Von Frey Filaments

Measurements of the paw withdrawal threshold and the 50% paw withdrawal threshold were significantly lower in the group of OVX mice treated with vehicle (OVX) than in the group of sham-operated mice treated with vehicle (SHAM). Treatment with CTLA-4Ig significantly improved pain behavior induced by OVX in both assays measuring paw withdrawal threshold and 50% paw withdrawal threshold (Figure 1A,B). Paw withdrawal frequencies following stimulations with 0.4–1.4 g filaments were significantly higher for the OVX group than for the SHAM group. Paw withdrawal frequencies in response to stimulations with 0.4–1.4 g filaments were lower in the CTLA-4Ig group than in the OVX group (Figure 1C).

### 2.2. Analysis of Three-Dimensional Bone Structure by μCT

Three-dimensional images of the distal femoral metaphyses (Figure 2A) and proximal tibial metaphyses (Figure 2B) showed less cancellous bone in the OVX group than in the SHAM group. Cancellous bone loss was lower in the CTLA-4Ig group than in the OVX group in the distal femoral metaphyses and proximal tibial metaphyses.

Micro-computed tomography (μCT) analysis of the distal femoral metaphyses and proximal tibial metaphyses showed that bone volume/tissue volume (BV/TV) and trabecular number (Tb.N) were significantly lower in the OVX group than in the SHAM group, whereas trabecular separation (Tb.Sp) was significantly higher in the OVX group compared to the SHAM group. Thus, OVX induced significant osteoporotic changes detected by μCT analysis of the knee. Interestingly, the analysis of samples from CTLA-4Ig-treated mice showed significant improvements in the parameters BV/TV, Tb.N, and Tb.Sp in the proximal tibial metaphyses, and CTLA-4Ig administration tended to improve also the bone structure in the distal femoral metaphyses (Figure 2C–H). There was no significant difference in trabecular thickness (Tb.Th) between the distal femur and the proximal tibia among all groups (Figure 2I,J).

### 2.3. Histological Analysis

The OVX group showed less cancellous bone in the distal femoral metaphyses and proximal tibial metaphyses than the SHAM group. The CTLA-4Ig group showed improvement in cancellous bone loss compared to the OVX group (Figure 3A). The number of tartrate-resistant acid phosphatase (TRAP)-positive osteoclasts in the distal femoral metaphyses and proximal tibial metaphyses was significantly higher in the OVX group than in the SHAM group, whereas it was significantly lower in the CTLA-4Ig group in comparison to the OVX group (Figure 3A–D). Thus, treatment with CTLA-4Ig suppressed TRAP activity.

### 2.4. Effect of Hind Limb Unloading on mRNA Levels of TNF-α, Wnt-10b, and SOST

The messenger RNA (mRNA) levels of TNF-α and sclerostin (SOST) in the hind limb bone were upregulated by OVX (relative expression vs. control) compared with the mRNA levels in SHAM group; however, this difference was not statistically significant. The mRNA levels of TNF-α and SOST tended to decrease with CTLA-4Ig treatment compared to those without any treatment; relative expression in the OVX group vs. that in the SHAM group was: TNF-α, 4.226; SOST, 1.944; relative expression in the CTLA-4Ig group vs. the SHAM group was: TNF-α, 1.686; SOST, 0.395 (Appendix A). The mRNA levels of Wnt-10b were downregulated by OVX compared to SHAM and CTLA-4Ig treatments (relative expression of Wnt-10b, in the OVX group vs. the SHAM group was 0.269; relative expression of Wnt-10b, in the CTLA-4Ig group vs. the SHAM group was 1.176) (Appendix A).

## 3. Discussion

Osteoporosis is a common disorder of the skeleton characterized by impairment of the fine balance between osteoclast bone resorption and osteoblast bone formation, conditions that predispose to bone loss [10]. Osteoporosis, even without fractures, has been associated with severe discomfort and/or disability and affects different aspects of personal life, with a variety of undesirable consequences, such as chronic pain, reduced physical ability, reduced social activity, and depressed mood [11]. Bone pain is one of the most common complications in cancer patients with bone metastases. Previous studies of bone pain in a metastatic cancer model have shown that the acidic microenvironment created by bone-resorbing osteoclasts activates transient receptor potential channels of the vanilloid subfamily member 1 (TRPV1) [12]. Our previous studies using an animal model of OVX-induced osteoporotic pain suggest that treatment of osteoporosis is useful for osteoporotic pain associated with osteoclast activity and TRPV1 stimulation [13,14].

The immune system has powerful effects on bone turnover. Physiologically, B cells secrete osteoprotegerin, a potent anti-osteoclastogenic factor that preserves bone mass [10]. Estrogen (E2) deficiency leads to bone loss through a complex cascade of interacting pathways involving the immunoskeletal interface. Estrogen is known to mediate potent anti-inflammatory effects in the body, and loss of estrogen has been shown to cause significant expansion of lymphocytes, both T cells [15] and B cells [16]. These studies suggested a model in which estrogen depletion results in the expansion of T cells that secrete TNF-α, and this TNF-α amplifies receptor activator of nuclear factor-kappa B ligand (RANKL)-induced osteoclastic bone resorption causing bone loss [17]. In our study, OVX increased the mRNA level of TNF-α, as well as the number of TRAP-positive osteoclasts in the hindlimb bone.

Interestingly, in estrogen deficiency, activated T cells secrete RANKL, TNF-α, and interleukin (IL)-17A, which amplify bone resorption and contribute to postmenopausal osteoporosis [10]. IL-17A and TNF-α also promote bone loss in inflammatory states such as RA [10]. Several studies attest to the critical role inflammation plays in the development of systemic osteoporosis in RA. For example, cytokines such as TNF-α, IL-6, IL-1β, and immune cell-derived RANKL have a detrimental effect on osteoblastogenesis and a positive effect on osteoclastogenesis [18]. The study by Laan et al. was the first to show with dual-energy X-ray absorptiometry (DXA) that RA patients have lower BMD values compared to matched controls [19]. Gough et al. demonstrated an accelerated BMD loss in early RA patients in comparison to controls at the spine and trochanter [20].

The introduction of bDMARDs for the treatment of RA allowed not only a reduction in cartilage damage but also a decrease in both localized and generalized bone loss. Several studies in RA have reported beneficial effects on bone mass after treatment with bDMARDs [5]. In a clinical one-year prospective open-label study, patients with active RA receiving tocilizumab, an anti-IL-6 receptor antibody, had no change in BMD and presented an increase in a bone formation marker [21]. However, only RA patients with osteopenia at baseline had significantly increased BMD values of the lumbar spine and femoral neck.

Our recent study reported that an anti-IL-6 receptor antibody treatment prevented ovariectomy-induced or unloaded-induced mechanical hyperalgesia in the hindlimbs, but the treatment had no effect on the induced bone loss [22,23].

In the present study, CTLA-4Ig treatment decreased mechanical hyperalgesia in an ovariectomized osteoporotic murine model with the prevention of bone loss and the tendency to inhibit the release of inflammatory cytokines. Clinically, ABT increases BMD at the femoral neck and maintains it at the lumbar spine, offering comparable efficacy to other bDMARDs [9].

Bone formation is promoted when Wnt-10b binds to the Wnt receptors low-density lipoprotein receptor-related protein 5 (LRP-5) and LRP-6 on osteoblasts. Wnt-10b expression is upregulated in anergic T cells [24]. Similarly, our results showed that the mRNA level of Wnt-10b was decreased, and the mRNA level of SOST was increased in OVX mice compared to non-OVX mice. The CD28 receptor on T cells associates with CD80/CD86 ligands on antigen-presenting cells and mediates key costimulatory signals necessary for T cell activation, downstream of the T cell receptor engagement of antigens on antigen-presenting cells [25]. CTLA-4, which has a strong affinity for CD80/CD86, binds to CD80/CD86 and suppresses T cells [24]. CTLA-4Ig prevents bone loss in OVX mice [26]. In addition, ABT, a soluble fusion protein formed by the extracellular domain of human CTLA4 linked to a human IgG1 Fc portion, improved in our study the mRNA levels of Wnt-10b and SOST in OVX mice. However, in our data, the change in Wnt10b expression was not statistically significant. In support of our findings, CTLA-4 promoted Wnt-10b production and bone formation under physiological conditions in mice also in a previous study [24]. CTLA-4 has recently been reported to inhibit osteoclast differentiation by inducing the indoleamine 2,3-dioxygenase/tryptophan pathway [27]. Thus, ABT may be effective not only for suppressing bone resorption but also for improving bone formation.

Maggi et al. reported that ABT reduces the proliferative response to recall antigens and the production of proinflammatory cytokines such as interferon (IFN)-γ and TNF-α in healthy donors in vitro. [28]. In an in vivo study, CTLA-4Ig suppressed the generation of large amounts of TNF-α and IFN-γ by splenocytes from DBA/1 mice with glucose-6-phosphate isomerase-induced arthritis [29]. These research reports support our result that CTLA-4Ig in vivo treatment suppresses the expression of TNF-α mRNA in vivo.

The present study has several limitations. First, we did not evaluate biomarkers related to pain, such as calcitonin gene-related peptide and TRPV1. Second, the dose- and time-dependent effects of CTLA-4Ig were not examined. Third, whether CTLA-4Ig administration prior to disease onset prevents osteoporosis is yet to be investigated. Fourth, although we observed changes in gene expression of TNFα, SOST, and Wnt10b with CTLA-4Ig, none of the data were statistically significant. This indicates weak support for an altered cytokine profile in the OVX group following CTLA-4Ig treatment. Future studies should address the timing effects of CTLA-4Ig treatment in OVX-induced bone loss. Furthermore, studies should examine inflammatory cytokines (IFN-γ, TNF-α, etc.) at the systemic and local levels. Using enzyme-linked immunosorbent assays, these studies should examine how CTLA-4Ig treatment alters systemically the serum cytokine profile, and the immunohistochemical expression of several inflammatory cytokines in the hind limb should be evaluated as a local marker at 4 weeks after treatment.

## 4. Materials and Methods

### 4.1. Reagents

The reagent used was CTLA-4Ig (Orencia^®^ for intravenous infusion; Bristol-Myers Squibb, Tokyo, Japan).

### 4.2. Animals

The experiment was approved by the Mie University Animal Care Committee (approval number: 2721, approval date 4 January 2016) and was undertaken in accordance with the ethical guidelines of the National Institutes of Health. A randomized, prospective, controlled animal model design was used. All efforts were made to minimize animal suffering and the number of animals used. Sample size was determined using a power analysis for an alpha of 0.05 and a power of 0.80 using G*POWER3 [30].

Seven-week-old female ddY mice were purchased from Japan SLC (Hamamatsu, Shizuoka, Japan) and acclimated for 1 week before the start of the experiments. Two mice were housed per cage in a temperature-controlled room (23 ± 1 °C) with a 12 h light/dark cycle (lights on from 7:00 to 19:00) and were given free access to food and water.

### 4.3. Experimental Protocol

At 8 weeks of age, the mice were randomly assigned to three groups (*n* = 6/group): sham-operated mice treated with vehicle (SHAM), OVX mice treated with vehicle (OVX), and OVX mice treated with CTLA-4Ig (CTLA-4Ig). The mice were either ovariectomized bilaterally under anesthesia with pentobarbital sodium (Sankyo, Tokyo, Japan) administration intraperitoneally or sham-operated (ovaries exteriorized but not removed). Starting immediately after surgery, vehicle or 25 mg/kg CTLA-4Ig was injected intraperitoneally three times a week for 4 weeks [25]. At the end of the 4-week treatment period, the mechanical sensitivity of the hind limbs was tested using von Frey filaments. Following the test, mice were sacrificed with an intraperitoneal injection of pentobarbital sodium (0.5 mg/kg). The bilateral hind limbs were removed for μCT, immunohistochemical analysis, and mRNA expression analysis.

### 4.4. Measurement of Pain-Related Behavior with Von Frey Filaments

Mechanical hyperalgesia around knee osteoporosis was assessed with the von Frey filaments. The von Frey test was conducted after drug or vehicle administration for 4 weeks, as described previously [14,22,23]. The frequency of the withdrawal response was evaluated that five von Frey filaments with forces of 0.4 g, 0.6 g, 1.0 g, 1.4 g, and 2.0 g were applied five times each in ascending order of force. The results were expressed as the percent response frequency of paw withdrawals. The withdrawal threshold was evaluated that each von Frey filament was applied once, starting at 0.008 g, with increasing force until a withdrawal response was reached, which was considered a positive response. The lowest force producing a response was considered the withdrawal threshold. The 50% withdrawal threshold was evaluated that a series of nine von Frey filaments, calibrated to produce incremental forces of 0.02 g, 0.04 g, 0.07 g, 0.16 g, 0.4 g, 0.6 g, 1.0 g, 1.4 g, and 2.0 g, were applied. Testing was initiated with a 0.6 g filament. Testing was initiated with a 0.6 g filament. In the absence of a clear paw withdrawal response, increasingly stronger filaments were presented consecutively until one of them was found to elicit a positive response. If the 0.6 g filament elicited a response, filaments with decreasing strength were presented until the identification of the first filament that failed to cause paw withdrawal. Data were collected using the up–down method [31] to calculate the 50% mechanical paw withdrawal threshold.

### 4.5. Analysis of Three-Dimensional Bone Structure by μCT

To determine the three-dimensional bone structure, isolated femurs and tibias were imaged using a μCT scanner (R_mCT; Rigaku Corporation, Tokyo, Japan), as described previously [13,14,22,23]. The scanned region contained both cortical and trabecular bone in the distal femoral metaphysis and the proximal tibial metaphysis located approximately 200 μm from the growth plate. Three-dimensional images were reconstructed and analyzed using a three-dimensional image analysis software (TRI/3D-BONE; RATOC System Engineering, Tokyo, Japan). The bone structure was evaluated based on the parameters bone volume fraction (BV/TV, %), Tb.N (/mm), Tb.Th (μm), and Tb.Sp (μm).

### 4.6. Histological Analysis of the Hind Limb Bone

Isolated hind limb bones and immunostained sections were prepared as described previously [13,14,22,23]. Right hind limb bones were fixed in 4% paraformaldehyde for 1 day. Tibias were decalcified in 10% EDTA for 2 weeks. After embedding in paraffin, sections were stained with hematoxylin and eosin for histological analysis of the bone structure. To identify osteoclasts in the hind limb bone, the TRAP method was used. In the proximal tibia, the number of TRAP-positive osteoclasts was determined within an area of 0.5 × 2 mm (length × width), apart from the most distal part of the growth plate. The immunostained sections were reviewed by different observers who were blinded to the experimental group.

### 4.7. RNA Isolation

Total cellular RNA was extracted from the left hind limb bone of the mice using Isogen (Nippon Gene, Toyama, Japan), according to the manufacturer’s instructions. Total RNA was reverse-transcribed using the Transcriptor First Strand cDNA Synthesis kit (Roche Applied Science, Penzberg, Germany) with a DNA thermal cycler (Veriti; Applied Biosystems, Foster City, CA, USA), according to the manufacturer’s protocol.

### 4.8. Quantitative Real-Time Polymerase Chain Reaction

The resultant complementary DNA (in triplicate) was amplified for the genes coding for the following molecules: TNF-α, Wnt-10b, and SOST. Inventoried (ready-made) primers corresponding to the target genes were used in this study (Table 1; TaqMan Gene Expression Assays; Applied Biosystems). Real-time PCR was performed using the ABI PRISM 7000 Sequence Detection System (Applied Biosystems). PCRs were carried out in duplicate with 1 cycle at 50 °C for 2 min, 1 cycle at 95 °C for 10 min, and 40 cycles at 95 °C for 15 s and 60 °C for 1 min. The assay was calibrated using glyceraldehyde 3-phosphate dehydrogenase as an internal control.

### 4.9. Statistical Analysis

Correlations among the SHAM, OVX, and CTLA-4Ig groups were tested using one-way ANOVA followed by the Bonferroni multiple comparison test if data were normally distributed. If data were not distributed normally, they were analyzed using the Kruskal–Wallis test; *p* < 0.05 was considered significant. All statistical analyses were performed using IBM SPSS Statistics 26 (IBM Japan, Tokyo, Japan).

## 5. Conclusions

In summary, CTLA-4Ig partially prevented bone loss and improved mechanical hyperalgesia in the hind limbs of OVX mice. The preventive mechanism may involve the improvement of TNF-α, Wnt-10b, and SOST expression, as well as osteoclast functions. The results of this study enhance our understanding of osteoporotic pain and suggest that CTLA-4Ig might preserve bone health and decrease osteoporotic pain.

## Figures and Tables

**Figure 1 ijms-21-09479-f001:**
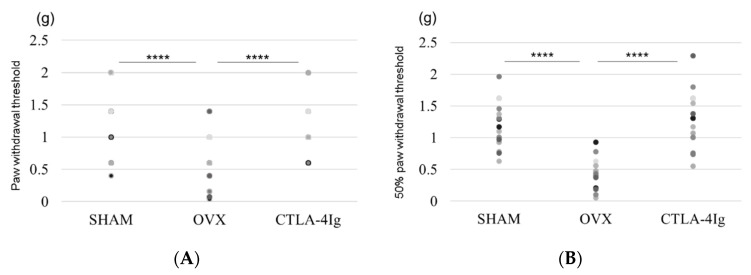
Measurement of pain-related behavior with von Frey filaments. (**A**) Paw withdrawal threshold; (**B**) 50% paw withdrawal threshold by the up–down method; (**C**) withdrawal frequency stimulation. Data are shown as scatter plots (* *p* < 0.05, ** *p* < 0.01, **** *p* < 0.001; *n* = 12 in each group). SHAM, sham-operated mice, OVX, ovariectomy-treated mice, CTLA-4Ig, cytotoxic T lymphocyte-associated antigen-4 immunoglobulin G.

**Figure 2 ijms-21-09479-f002:**
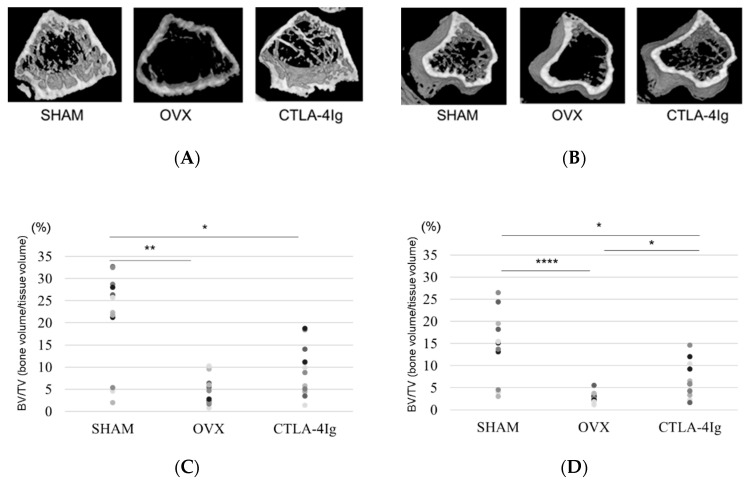
Micro-CT analyses of the distal femoral metaphysis and the proximal tibial metaphysis. Three-dimensional images of the distal femoral metaphysis (**A**) and the proximal tibial metaphysis (**B**); (**C**) bone volume/tissue volume (BV/TV, %) (distal femoral metaphysis); (**D**) BV/TV (%) (proximal tibial metaphysis); (**E**) trabecular number (Tb.N, per mm) (distal femoral metaphysis); (**F**): Tb.N (/mm) (proximal tibial metaphysis); (**G**) trabecular separation (Tb.Sp, μm) (distal femoral metaphysis); (**H**) Tb.Sp (μm) (proximal tibial metaphysis); (**I**): trabecular thickness (Tb.Th, μm) (distal femoral metaphysis); (**J**) Tb.Th (μm) (proximal tibial metaphysis). Data are shown as scatter plots (* *p* < 0.05, ** *p* < 0.01, *** *p* < 0.005, **** *p* < 0.001; *n* = 12 in each group).

**Figure 3 ijms-21-09479-f003:**
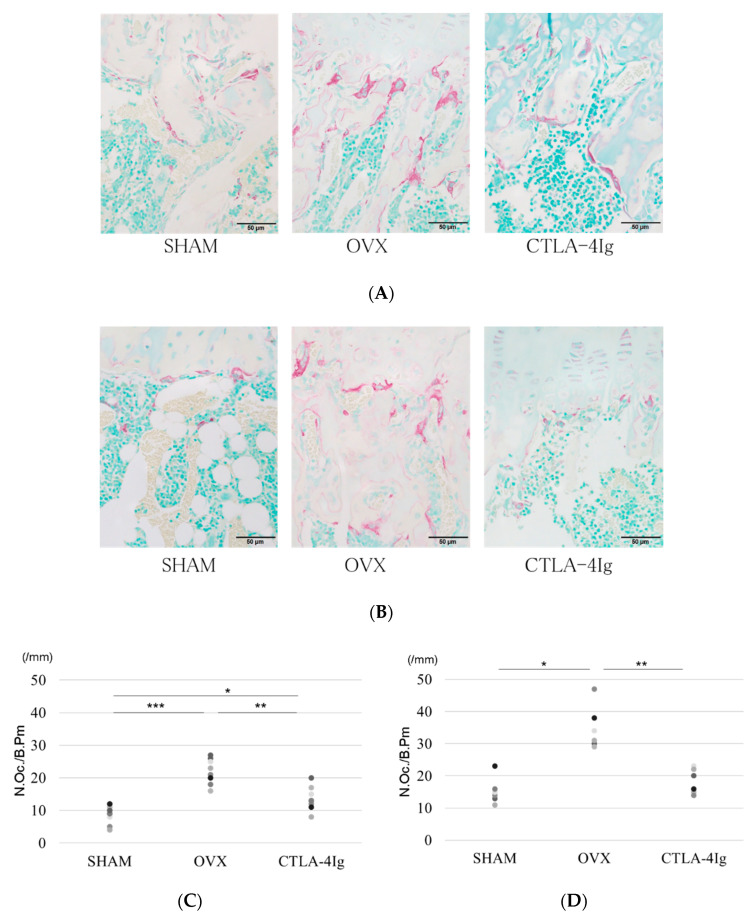
Histological analysis of hindlimb bone. (**A**) Tartrate-resistant acid phosphatase (TRAP) staining for the histological examination of the distal femoral metaphysis scale bar is 50 μm); (**B**) TRAP staining for the histological examination of the proximal tibial metaphysis; (**C**) histological analysis of the number of TRAP-positive osteoclasts in the distal femoral metaphysis; (**D**) histological analysis of the number of TRAP-positive osteoclasts in the proximal tibial metaphysis; (* *p* < 0.05, ** *p* < 0.01, *** *p* < 0.005; *n* = 12 in each group). N.Oc./B.Pm.: number of osteoclasts/bone perimeter.

**Table 1 ijms-21-09479-t001:** Primers for real-time polymerase chain reaction (PCR).

Genes	Assay ID ^a^	Size (bp)
TNF-α	Mm00443258_m1	81
SOST	Mm00470479_m1	55
Wnt-10b	Mm00442104_m1	57
GAPDH	Mm99999915_g1	109

**^a^** TNF-α Tumor Necrosis Factor-α, SOST screlostin, GAPDH Glyceraldehyde-3-phosphate dehydrogenase. TaqMan Gene Expression Assays (Applied Biosystems).

## Data Availability

The datasets used and/or analyzed during the study are available from the corresponding author upon request.

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
