# Peer review of "CTLA-4Ig Improves Hyperalgesia in a Mouse Model of Osteoporosis"

_ijms, 2020, doi:10.3390/ijms21249479_

Round 1
Reviewer 1 Report
CTLA-4Ig improves hyperalgesia in a mouse model of osteoporosis
Recommendation: major revision
In the manuscript the authors present a study done on ovariectomized mice to induce osteoporosis. In the context of bone structure remodeling, pain-related behavior and gene expression of bone formation-related markers, the animals were treated with a dosage of 25 mg/kg CTLA-4Ig intraperitoneally to improve the tested parameters. They observed, in the case of CTLA-4Ig administration, an improvement of bone loss related processes, when osteoporosis related processes were induced by ovariectomy.
The authors should improve Fig2 A OVX, and choose a representative image with better resolution (voxel size). The shown image makes a blurry impression, which make it hard to identify trabecular structures.
How did the authors quantify the number of osteoclasts assessed in their histological examinations?
If histological images with higher resolution are available, the authors should consider to replace the current ones.
The authors should introduce a graph displaying observed mRNA levels, in order to improve the assessment of the mentioned data. In the way the data is currently presented it is very confusing.
Please state how many animals per treatment group were prepared for histological examination.
Please state how many animals per treatment group received RNA isolation for gene expression analysis.
The authors should state the version of IBM SPSS used in the statistics.
Author Response
Replies to Reviewer #1
Authors are grateful to Reviewer #1 for encouraging comments. We have revised the indicated parts of the manuscript according to the comments. Corrections in the newly revised manuscript are underlined.
Please note that your review comments are shown in italic below and our replies in non-italic.
On the comment of [The authors should improve Fig2 A OVX, and choose a representative image with better resolution (voxel size). The shown image makes a blurry impression, which make it hard to identify trabecular structures.
If histological images with higher resolution are available, the authors should consider to replace the current ones.]
Reply: Thank you for your valuable pointing and suggestions. We provided the higher magnification of groups in TRAP expression in the hindlimb bones.
On the comment of [How did the authors quantify the number of osteoclasts assessed in their histological examinations?]
Reply: Thank you for suitable comment. We noted that “the number of TRAP-positive osteoclast was determined within an area 0.5 mm in length and 2 mm in width” on Line264-267. The three investigators performed the blinded analysis. The number of reactive TRAP cells were counted and measured by each one observer who was blinded to the experimental group. We added the following sentences, “The immunostained sections were reviewed by each one observer who was blinded to the experimental group” on Line273-274.
On the comment of [The authors should introduce a graph displaying observed mRNA levels, in order to improve the assessment of the mentioned data. In the way the data is currently presented it is very confusing.]
Reply: Thank you for your valuable suggestions. We added the graph displaying observed mRNA levels as Supplementary data.
On the comment of [Please state how many animals per treatment group were prepared for histological examination.
Please state how many animals per treatment group received RNA isolation for gene expression analysis.]
Reply: Thank you for your valuable suggestions. We noted the number of animals in each figure legend, “n = 12 in each group”.
On the comment of [The authors should state the version of IBM SPSS used in the statistics.]
Reply: Thank you for suitable comment and suggestion. We added “BM SPSS Statistics 26” as the version of IBM SPSS on Line294
Reviewer 2 Report
In this manuscript, Nagao at al examine the effect of CTLA-4Ig treatment a well established mouse model of osteoporosis. This is a well-written and designed study to determine if CTLA-4Ig treatment alters the outcome (pain, bone loss, and IHC) in OVX mice. The data are clear and mostly support the conclusions presented by the authors. I have several minor suggestions and several major issues that need to be addressed prior to publication. My major issues are with the interpretation and discussion of the qPCR data. Please see the comments below.
Minor suggestion:
Line 23. Should be a comma not a period after ‘examined’.
Figure 2. Please move the lines denoting statistical significance so they do not overlap with the y-axis values (G and H).
Figure 3. Please label what the y-axis abbreviation (N.Oc/B.Pm) represents in the figure legend.
Major suggestions:
Graphs should be presented as scatter plots rather than box plots so the individual data points can be observed.
The authors state in the discussion that ‘In the present study, CTLA-4Ig treatment decreased mechanical hyperalgesia in an ovariectomized osteoporotic murine model with prevention of bone loss and inhibition of inflammatory cytokines.’ While I agree with most of this statement, there is no data to support that CTLA-4Ig treatment altered the cytokine profile in the OVX group.
I believe the authors are basing this statement off the non-significant change in mRNA expression of tnf-a. In lines 129-138 the authors examine expression of tnf-a, sost and wnt10b. While they state there is a change in gene expression with CTLA-4Ig, none of the data is statistically significant.
These changes are not significant and cannot be used to infer that CTLA-4Ig treatment alters gene expression in any manner. This needs to be corrected in the manuscript. This data should be presented as a figure.
It is also not clear if the change in wnt10b expression is statistically significant. Given CTLA-4Ig treatment is should reduce proinflammatory cytokines, this is important to demonstrate if the authors want to make any conclusions about inflammatory cytokines or discuss any potential mechanisms related to their data and inflammatory signaling in the discussion.
This data (and resultant discussion) could be removed from the manuscript, but I have several suggestions that could improve the manuscript and the study.
The authors should examine either systemic or local inflammatory cytokines. This should be done using ELISAs (either a multiplex panel of several individual ones) to determine if CTLA-4Ig treatment alters the cytokine profile. The authors could collect serum each week for 4 weeks to examine how CTLA-4Ig treatment alters the cytokine profile or they could collect hind limbs at 4 weeks post treatment for a terminal analysis. While the local analysis would be more informative, it might be difficult to detect changes between experimental groups. However, this experiment is performed and whatever the outcome, it would be interesting to see the results.
Alternatively, the authors could examine the immunohistochemical expression of several inflammatory cytokines (INF-g, TNF-a, etc) in the hind limb.
While the quantification of inflammatory cytokines would be the most informative for this study, the authors may examine a larger selection of transcripts. This could be done with the RNA that was used for qPCR. While whole transcriptome analysis might not be necessary, there are many targeted platforms that could be used (e.g., Nanostring Inflammatory Panel). If the authors can provide quantification of inflammatory cytokines, this data would not be necessary.
Author Response
Replies to Reviewer #2
The authors are grateful to Reviewer #2 for insightful and critical review comments that significantly help improve the manuscript and increase clinical relevance. Corrections in the newly revised manuscript are underlined.
Please note that your review comments are shown in italic below and our replies in non-italic.
Minor suggestion:
On the comment of [Line 23. Should be a comma not a period after ‘examined’.]
Reply: Thank you for suggestion of the mistake. We changed a comma after ‘examined’ on Line23. We changed “After 4 weeks of treatment, mechanical sensitivity was examined, and the bilateral hind limbs were removed and evaluated for micro-computed tomography, immunohistochemical analyses, and messenger RNA expression analysis.”
On the comment of [Figure 2. Please move the lines denoting statistical significance so they do not overlap with the y-axis values (G and H).]
Reply: Thank you for suitable comment and suggestion. We moved the lines denoting statistical significance of all Figures.
On the comment of [Figure 3. Please label what the y-axis abbreviation (N.Oc/B.Pm) represents in the figure legend.]
Reply: Thank you for suitable comment and suggestion. This means the number of osteoclasts/bone perimeter (N.Oc./B.Pm.). We added the abbreviation (N.Oc/B.Pm) represents in the figure legend on Line126 and Abbreviations.
Major suggestions:
On the comment of [Graphs should be presented as scatter plots rather than box plots so the individual data points can be observed.]
Reply: Thank you for suitable comment and suggestion. However, we can not display scatter plots graph in withdrawal frequency stimulation and messenger RNA expression. We changed that figures 1-3 were expressed as scatter plots.
With Excel software, the same plot number overlapped, so the number of some groups look small in figure1C, withdrawal frequency stimulation.
On the comment of [The authors state in the discussion that ‘In the present study, CTLA-4Ig treatment decreased mechanical hyperalgesia in an ovariectomized osteoporotic murine model with prevention of bone loss and inhibition of inflammatory cytokines.’ While I agree with most of this statement, there is no data to support that CTLA-4Ig treatment altered the cytokine profile in the OVX group.
I believe the authors are basing this statement off the non-significant change in mRNA expression of tnf-a. In lines 129-138 the authors examine expression of tnf-a, sost and wnt10b. While they state there is a change in gene expression with CTLA-4Ig, none of the data is statistically significant.
These changes are not significant and cannot be used to infer that CTLA-4Ig treatment alters gene expression in any manner. This needs to be corrected in the manuscript.]
Reply: Thank you for suitable comment. We changed “In the present study, CTLA-4Ig treatment decreased mechanical hyperalgesia in an ovariectomized osteoporotic murine model with prevention of bone loss and inhibitory tendency of inflammatory cytokines.”on Line 177-179
On the comment of [It is also not clear if the change in wnt10b expression is statistically significant. Given CTLA-4Ig treatment is should reduce proinflammatory cytokines, this is important to demonstrate if the authors want to make any conclusions about inflammatory cytokines or discuss any potential mechanisms related to their data and inflammatory signaling in the discussion.]
Reply: Thank you for suitable comment and suggestion. As you pointed out, our data were not significantly different. We cited Ref 24 for Wnt-10b, but we added the following sentence,”in our data, the change in wnt10b expression was not statistically significant” on Line 191
On the comment of [This data should be presented as a figure.]
Reply: Thank you for suitable suggestion. However, the change in mRNA expression was not statistically significant. We added the graph displaying observed mRNA levels as Supplementary data.
On the comment of [This data (and resultant discussion) could be removed from the manuscript, but I have several suggestions that could improve the manuscript and the study. The authors should examine either systemic or local inflammatory cytokines. This should be done using ELISAs (either a multiplex panel of several individual ones) to determine if CTLA-4Ig treatment alters the cytokine profile. The authors could collect serum each week for 4 weeks to examine how CTLA-4Ig treatment alters the cytokine profile or they could collect hind limbs at 4 weeks post treatment for a terminal analysis. While the local analysis would be more informative, it might be difficult to detect changes between experimental groups. However, this experiment is performed and whatever the outcome, it would be interesting to see the results.
Alternatively, the authors could examine the immunohistochemical expression of several inflammatory cytokines (INF-g, TNF-a, etc) in the hind limb.]
Reply: Thank you for suitable comment and suggestion.
However, there is no blood or tissue sample to perform suggestion and there is not enough time to re-experiment. Therefore, we will describe this proposal as a limitation.
We added the following sentences.
“While there is a change in gene expression of TNFα, SOST and Wnt10b with CTLA-4Ig, none of the data was statistically significant. So, there is weak to support that CTLA-4Ig treatment altered the cytokine profile in the OVX group” on Line205-207.
“future studies should examine either systemic or local inflammatory cytokines. The studies should examine the serum cytokines (INF, TNF, etc) using ELISAs how CTLA-4Ig treatment alters systemically the cytokine profile, and the immunohistochemical expression of several inflammatory cytokines in the hind limb as local, at 4 weeks post treatment” on Line209-212.
Round 2
Reviewer 1 Report
The authors adjusted the manuscript accordingly
Reviewer 2 Report
The authors have addressed all my concerns and the manuscript should be accepted for publication.